# Design, Synthesis and Anticancer Profile of New 4-(1*H*-benzo[*d*]imidazol-1-yl)pyrimidin-2-amine-Linked Sulfonamide Derivatives with V600EBRAF Inhibitory Effect

**DOI:** 10.3390/ijms221910491

**Published:** 2021-09-28

**Authors:** Mohammed S. Abdel-Maksoud, Ahmed A. B. Mohamed, Rasha M. Hassan, Mohamed A. Abdelgawad, Garri Chilingaryan, Samy Selim, Mohamed S. Abdel-Bakky, Mohammad M. Al-Sanea

**Affiliations:** 1Pharmaceutical and Drug Industries Research Division, Medicinal & Pharmaceutical Chemistry Department, National Research Centre (ID: 60014618), Dokki, Giza 12622, Egypt; rashahassan_pharma@yahoo.ca; 2Department of Medicinal Chemistry, Faculty of Pharmacy, Mansoura University, Mansoura 35516, Egypt; ahmed_smt@yahoo.com; 3Department of Pharmaceutical Chemistry, College of Pharmacy, Jouf University, Sakaka 72341, Saudi Arabia; mhmdgwd@ju.edu.sa; 4Institute of Molecular Biology of NAS, Yerevan 0014, Armenia; g_chilingaryan@mb.sci.am; 5Department of Clinical Laboratory Sciences, College of Applied Medical Sciences, Jouf University, Sakaka 72341, Saudi Arabia; sabdulsalam@ju.edu.sa; 6Department of Pharmacology and Toxicology, College of Pharmacy, Qassim University, Buraydah 51452, Saudi Arabia; m.abdelbakky@qu.edu.sa

**Keywords:** anticancer, BRAF inhibitors, synthesis, benzimidazole, sulfonamide, cell cycle, virtual docking

## Abstract

A new series of 4-(1*H*-benzo[*d*]imidazol-1-yl)pyrimidin-2-amine linked sulfonamide derivatives **12a–n** was designed and synthesized according to the structure of well-established V600EBRAF inhibitors. The terminal sulfonamide moiety was linked to the pyrimidine ring via either ethylamine or propylamine bridge. The designed series was tested at fixed concentration (1 µM) against V600EBRAF, finding that **12e**, **12i** and **12l** exhibited the strongest inhibitory activity among all target compounds and **12l** had the lowest IC_50_ of 0.49 µM. They were further screened on NCI 60 cancer cell lines to reveal that **12e** showed the most significant growth inhibition against multiple cancer cell lines. Therefore, cell cycle analysis of **12e** was conducted to investigate the effect on cell cycle progression. Finally, virtual docking studies was performed to gain insights for the plausible binding modes of vemurafenib, **12i**, **12e** and **12l**.

## 1. Introduction

Cancer refers to a number of diseases characterized by the development of uncontrollable cells that can infiltrate and destroy normal body tissue. It is considered the second major cause of death after cardiovascular diseases [1,2]. In terms of 2020 statistics, it is the main cause of death in 112 out of 183 countries [3]. Traditional chemotherapeutics have severe and lethal side effects and have diminished efficacy owing to drug resistance [4,5]. Therefore, there is urgent need to explore new potent drugs with fewer side effects and higher efficacy.

The MAPK cascade, which includes RAS (Rat sarcoma virus kinase), RAF(Rapidly Accelerated Fibrosarcoma kinase), ERK (Extracellular signal-regulated kinase), and MEK (mitogen-activated protein kinase kinase) kinases, is considered one of the vital signaling pathways that controls cell function and proliferation [6,7,8,9]. BRAF (a serine/threonine kinase) is the key activator promoter of MAPK signaling pathway. The hyperactivation of MAPK pathway due to oncogenic mutation of BRAF (Rapidly Accelerated Fibrosarcoma kinase isoform B) occurs in 70% of melanoma cases [10,11], 10% of colorectal cases [12], and 70% of thyroid cancer cases [13,14,15]. The main oncogenic mutation of BRAF occurs when the valine amino acid in position 600 is replaced by glutamic acid to produce V600EBRAF, which is the major mutation in melanomas [16,17,18,19,20] and papillary thyroid carcinomas [10], accounting for up to 63% in melanoma and 50% in papillary thyroid carcinomas. This, in turn, has drawn attention to the possibility of targeting V600EBRAF as a treatment for melanoma with hyperactive MAPK pathways [21,22,23].

Sorafenib (Nexavar^®^, BAY- 43-9006) was the first multikinase inhibitor to have a dual inhibitory activity against BRAF and CRAF (Rapidly Accelerated Fibrosarcoma kinase isoform C) [24,25]. Vemurafenib was the first selective V600EBRAF inhibitor with sulfonamide moiety to be approved for the treatment of melanoma [26]. In addition, PLX 4720, encorafenib, and dabrafenib showed selective V600EBRAF inhibitory activity for the treatment of late-stage melanoma via induction of programmed cell death (Figure 1) [27,28]. It is worth noting that melanoma without mutated BRAF was not treated by either vemurafenib or dabrafenib [25,29,30]. 

Benzimidazole is a heterocyclic compound composed of two fused rings of benzene and imidazole. It is a constitutional isomer for azaindole ring that is present in the vemurafenib family. It shows a vast range of therapeutic activities including anticancer [31,32,33,34], antiviral [35,36,37], antifungal [38,39], and anthelmintic effects [40,41,42].

In our previous work, different series of compounds containing sulfonamide moiety showed high inhibitory activity towards V600EBRAF, wild-type BRAF, and CRAF. Different backbone scaffolds were evaluated such as pyrazole, imidazole, imidazothiazole, imidazoxazole, and azaindole [43,44,45,46,47,48,49]. In the present work, pyrimidinylbenzimidazole scaffold was used as a main backbone for the new sulfonamide derivatives. The linker between sulfonamide moiety and pyrimidine is either ethylamino or propylamino bridge. The target compounds were screened for their inhibitory effect over V600EBRAF at fixed concentration. To check both potency and selectivity, we determined the IC_50_ of the target compounds over V600EBRAF, wild-type BRAF, and CRAF. The final compounds were subsequently submitted to NCI (National Cancer institution) to investigate their effect on different cancer cell lines. Moreover, cell cycle analysis for the most promising compound was performed.

## 2. Results and Discussion

### 2.1. Chemistry

Final compounds **12a–n** were prepared via three distinct steps. The first step includes the synthesis of main intermediate 1-(2-(methylsulfonyl) pyrimidin-4-yl)-1*H*-benzo[*d*]imidazole **(4)** starting from o-phenylenediamine **(1)** (Figure 1). The second step involves the synthesis of terminal side chain **11a–n** starting from ethanolamine **(5a)** and propanolamine **(5b)** (Figure 2). Finally, coupling of **4** and **11a–n** led to the formation of the target compounds **12a–n**.

As illustrated in Figure 1, heating a mixture of o-phenylenediamine **(1)** and 90% formic acid at 100 °C for 2h and then cooling the mixture followed by slow addition of 10% NaOH led to the precipitation of 1*H*-benzo[*d*]imidazole **(2)**. Arylation of **2** with 4-chloro-2-(methylthio) pyrimidine applying NaH as a base in DMF afforded 1-(2-(methylthio)pyrimidin-4-yl)-1*H*-benzo[*d*]imidazole **(3)**, which upon oxidation by oxone in a mixture of MeOH and H_2_O led to the formation of the key intermediate 1-(2-(methylsulfonyl)pyrimidin-4-yl)-1*H*-benzo[*d*]imidazole **(4)**.

Next, we turned our attention to the synthesis of the terminal side chains **11a–n**, as shown in Figure 2. It involves the reaction of ethanolamine **(5a)** or propanolamine **(5b)** with benzyl chloroformate using triethyl amine as a base in DCM to give N-protected ethanolamine **(6a)** or propanolamine **(6b)**. Reaction of **6** with methane sulfonyl chloride using trimethylamine as a base in DCM afforded 2-(((benzyloxy) carbonyl) amino) ethyl methane sulfonate **(7a)** or 2-(((benzyloxy) carbonyl) amino) propyl methane sulfonate **(7b)**. Replacement of the mesylate group by azide via heating **7** with sodium azide in DMSO gave benzyl(2-azidoethyl)carbamate **(8a)** or benzyl(2-azidopropyl)carbamate **(8b)**. Reduction of **8** by applying Staudinger reduction using triphenylphosphine and heating in MeOH led to the formation of benzyl(2-aminoethyl)carbamate **(9a)** or benzyl(2-aminopropyl)carbamate **(9b)**. Reaction of **9** with arylsulfonyl chloride in DCM and trimethyl amine afforded **10a–n**. Deprotection of **10** using Pd/C under H_2_ atmosphere in MeOH led to the formation of the terminal side chains **11a–n**. 

Finally, reaction of **4** with **11a–n** in DMSO in presence of diisopropyl ethyl amine afforded the target compounds **12a–n** in 52–81% isolated yield (Figure 3, Table 1). 

### 2.2. Biology

#### 2.2.1. Enzyme Assay

The target compounds **12a–n** were designed to be V600EBRAF inhibitors, and therefore the first screening stage was to detect their ability to inhibit V600EBRAF at a fixed concentration of 1 µM. All target compounds showed certain inhibitory effect on V600EBRAF (Table 2). Generally, **12h–n** (with propylamine linker between pyrimidine ring and sulfonamide terminal) showed higher percent inhibition compared to **12a–g** (with ethylamine linker between pyrimidine ring and sulfonamide terminal). However, it was found that **12m** (*p*-OMe substituted benzene sulfonamide terminal) has lower percent inhibition compared to its ethylamine analogue **12f**.

Regarding the ethylamine linker containing **12a–g**, **12e** with *p*-CF_3_ substituted benzene sulfonamide terminal had the highest percent inhibition with mean value of 98%, followed by **12g** with *p*-CH_3_ substituted benzene sulfonamide terminal with mean percent inhibition of 82%. Furthermore, *p*-halobenzene sulfonamide derivatives **12b–d** showed size-dependent activity; the compound with smaller size halide **12b** (*p*-F) showed the highest percent inhibition with mean value of 75%, followed by **12c** (*p*-Cl) with mean percent inhibition 71% and finally **12d** (*p*-Br) with mean percent inhibition of 69%. **12a** (unsubstituted) and **12f** (*p*-OMe) showed the lowest activities with mean percent inhibitions of 30% and 55%, respectively. 

On the other hand, **12h-n** showed higher percent inhibition compared to the ethylamine analogues **12a–g**. **12l** (*p*-CF_3_ sulfonamide terminal) exhibited the highest percent inhibition with mean value of 99%. Compounds with *p*-halobenzene sulfonamide terminal **12i-k** showed relatively high size dependent activity. Compound **12i** (*p*-F) showed the highest activity with mean percent inhibition of 98%, followed by **12j** (*p*-Cl) with mean percent inhibition of 88%, and finally **12k** (*p*-Br) with mean percent inhibition of 74%. Compounds **12h** (unsubstituted) and **12m** (*p*-OMe) had moderate activity with mean percent inhibitions of 50% and 51%, respectively. **11n** (naphthyl) showed the lowest activity, with mean percent inhibition of 41%.

Next, we focused on determination of the potency (IC_50_) of each compound over V600EBRAF and finding the selectivity for V600EBRAF over wild-type BRAF and CRAF. Most compounds showed relatively high potency towards V600EBRAF with submicro molar IC_50_ values (Table 3).

With respect to the ethylamine linker containing **12a–g**, concerning wild type BRAF, **12e** showed the lowest IC_50_ with mean value 1.25 µM, followed by **12c** and **12b** with IC_50_ 1.38 and 1.51 µM, respectively. **12g** exhibited comparable activity to **12b** with IC_50_ 1.53 µM. **12d** showed the lowest activity among the halogenated **12b–d** with IC_50_ 1.79 µM. **12a** showed the lowest activity with IC_50_ 3.25 µM. Regarding V600EBRAF, **12e** had the highest activity with IC_50_ 0.62 µM, followed by **12b** and **12g** with IC_50_ 0.74 µM and 0.79 µM, respectively. **12a** had the lowest activity with IC_50_ 1.91 µM. Concerning CRAF, **11e** exhibited the highest activity with IC_50_ 1.14 µM, followed by **12g** and **12c** with IC_50_ 1.48 µM and 1.49 µM, respectively. On the other hand, **12a** and **12d** exhibited the lowest activities with IC_50_ 2.10 µM and 2.98 µM, respectively.

Regarding the propylamine linker containing **12h–n** against wild type BRAF, **12l** had the highest activity with IC_50_ 0.94 µM, followed by **12j**, **12i**, **12m** and **12k** with IC_50_ 1.13 µM, 1.33 µM, 1.46 µM and 1.53 µM, respectively. **12h** (unsubstituted) and **12n** (naphthyl) exhibited the lowest activity with IC_50_ 2.32 µM and 3.54 µM, respectively. Concerning V600EBRAF, **12l** showed the highest activity with IC_50_ 0.49 µM, followed by **12i** and **12j** with IC_50_ 0.53 µM and 0.79 µM, respectively. **12k** and **12m** exhibited similar potency with IC_50_ 0.81 µM and 0.82 µM, respectively. **12n** showed the lowest activity with IC_50_ 1.28 µM. Finally, studying the activity of **12h–n** over CRAF revealed that **12l** exhibited the highest activity with IC_50_ 0.84 µM. **12i**, **12k** and **12m** showed almost similar potencies with IC_50_ 0.98 µM, 0.99 µM and 0.96 µM, respectively. **12h** exhibited the lowest activity with IC_50_ 2.14 µM.

From the above-mentioned results, we can build a structure–activity relationship for the final compounds over the three RAF isoforms. Compounds that carry propylamine linker showed higher activity compared to their ethylamine analogues. Moreover, electron withdrawing groups such as flouro-, chloro-, and triflouromethyl groups enhanced the activity over electron-donating groups. Compounds containing bulky groups as bromo and naphthyl exhibited lower activity compared to those containing small groups as methyl and flouro. The optimum activity was obtained with the moderate size triflouro methyl and propylamine linker.

#### 2.2.2. In Vitro Antiproliferative Activity 

All target compounds were submitted to the NCI, USA, for in vitro antiproliferative assay over a 60-cell line panel representing nine different types of cancer (non-small cell lung cancer, leukemia, colon cancer, ovarian cancer, CNS cancer, melanoma, renal cancer, breast cancer, and prostate cancer) (Figure 2). In general, the compounds with ethylamine linker had the highest mean percent inhibition (Figure 2A). **12e** exhibited the highest mean percent inhibition with a mean value of around 50%, followed by **12g** and **12h** with mean percent inhibition of 30%. 

The sub-panel analysis of NCI data revealed that **12e** showed the most significant growth inhibition over most of the tested cancer subtypes. Regarding leukemia cell lines, **12e**, **12i** and **12l** possessed the highest activity (60%, 50% and 46% mean percent inhibition). Concerning non-small cell lung cancer cell lines, **12e** showed 50% mean percent inhibition, followed by **12g** and **12l**, both with 30% mean percent inhibition. **12e** and **12l** were found to be the most active against colon cancer cell lines with 40% and 30% mean percent inhibition, respectively. **12e** exhibited the same inhibitory activity over CNS and melanoma cancer cell lines with 50% mean percent inhibition. Concerning ovarian cancer cell lines, **12e** showed the highest activity with 40% mean percent inhibition, followed by **12g**, **12i** and **12l** with mean percent inhibitions of around 25%. **12e** showed the highest inhibition over renal cancer cell lines with mean percent inhibition of 45%, followed by **12g** and **12l** with mean percent inhibition around 30%. Only **12e** showed a significant inhibitory activity over prostate cancer cell lines with mean percent inhibition of 40%, followed by **12g**, **12i**, **12l** and **12n** with mean percent inhibitions of around 20%. Finally, **12e** showed more than 60% inhibition over breast cancer cell lines, followed by **12g**, **12i** and **12l** with mean percent inhibitions of around 40%.

Map heat analysis of the target compounds **12a–n** (Appendix A) for leukemia cell lines showed that **12e** exhibited the highest inhibitory effect over MOLT-4, RPMI-8226, and SR with percent inhibitions of 55%, 55% and 65%, respectively. **12g** showed around 45% inhibition over MOLT-4, RPMI-8226, and SR cell lines. **12i** exhibited around 60% inhibition over K-562, RPMI-8226, and SR. **12k** showed 30% inhibition over SR cell line. **12l** and **12n** showed around 60% and 30% inhibition over K-562 and MOLT-4 cell lines, respectively. Regarding non-small cell lung cancer, **12i** showed around 30% inhibition over EKVX, NCI-H226 and NCI-H522. **12e** exhibited around 60% inhibition over EKVX cell line and 40% inhibition over HOP-92, NCI-H23 and NCI-H460. **12g** and **12l** showed around 40% inhibition over HOP-92 and NCI-H226. Concerning colon cancer, **12e** showed about 60% inhibition over HCT-116 and around 40% inhibition over KM12 and HCT-15. **12i** showed around 50% inhibition over KM12. **12l** exhibited more than 40% over HCT-15 and HT29. Regarding CNS cancer cell lines, **12e** showed more than 50% inhibition over SF-295, SNB-75 and U251. In the case of melanoma cell lines, **12e, 12g, 12h**, **12i, 12k, 12n** and **12l** showed 91%, 80%, 40%, 97%, 40%, 40% and 70% inhibition over SK-MEL-5, respectively. In the case of ovarian cancer cell lines, **12e** and **12i** showed the highest inhibitory activity over OVCAR-4 with 71% and 56% inhibition, respectively. **12e** also exhibited good inhibitory activity over NCI/ADR-RES cell line with 45% inhibition. Regarding renal cancer cell lines, **12e** showed around 60% inhibition over UO-31, RXF 393 and CAKI-1 cell lines. **12g** exhibited 48% inhibition over UO-31 cell line. **12l** showed strong inhibitory effect over UO-31, RXF 393 and CAKI-1 with 46%, 57% and 49% inhibition, respectively. **12e** was the only compound to show significant inhibition on a prostate cancer cell line with 55% inhibition over the PC-3 cell line. Finally, in the case of breast cancer, **12e** exhibited 77% and 74% inhibition over T-47D and MDA-MB-468, respectively. **12g** showed significant inhibitory effect on T-47D and MDA-MB-468 with percent inhibitions of 73% and 49%, respectively. **12h**, **12i**, **12k**, **12l** and **12n** showed high inhibitory activity over T-47D with 55%, 75%, 55%, 84% and 56% inhibition, respectively.

#### 2.2.3. Cell Cycle Analysis 

The compound **12e** that possess a broad spectrum of activity over different cell lines was picked to study its consequences on cell cycle phases at a dose of 5 µM, the closest concentration to the IC_50_ (5.35 µM) of the compound **12e** on the tested cell line on T-47D (one of the most sensitive cell lines to **12e**) (Figure 3). The distribution of cell cycle was determined by flow cytometric assay after using propidium iodide to stain the DNA content of the treated cells. After 72h of incubation, **12e** showed an increase in the S-phase from 14% in the control cells to 18% in the treated cells, and it increased the G1 phase from 29% to 31%. Meanwhile, the cells of the G2-phase decreased to 6% compared to 8% in the untreated cells. This indicates that the compound **12e** prevents cancer cells from starting DNA division, which is characteristic for kinases inhibitors. 

### 2.3. Molecular Docking

Molecular docking of **12e**, **12i** and **12l** compounds and vemurafenib (reference ligand) in the binding site of the V600EBRAF protein was performed in order to study the differences in the binding modes of these compounds and in their interaction patterns with the amino acid residues of the protein.

The binding poses of these compounds, together with the 2D diagrams of their interaction with the amino acid residues of the V600EBRAF protein’s binding site, are presented in Figure 4. The molecular docking studies demonstrated that the compounds **12i** and vemurafenib have visibly different binding modes, both to each other and to the compounds **12e** and **12l**.

The compounds **12e**, **12i**, **12l** and vemurafenib demonstrated the following docking mfScores: –200.05, –164.08, –175.4 and –213.07, respectively. According to the results of the MMPBSA re-scoring of the obtained complexes, the compounds **12e**, **12i**, **12l** and vemurafenib showed the following binding energies: –23.66, –26.36, –25.86 and −31.41 kcal/mol, respectively.

In terms of the results of the molecular docking studies, cysteine 532 amino acid residue of the V600EBRAF protein’s binding site is of special importance for the interaction, since all three selected compounds and reference ligand showed hydrogen bonding formation with it (Figure 5). A481, F583, I463, K483, L514, S535, V471 and W531 are important for the hydrophobic interaction of the three selected compounds and vemurafenib with the V600EBRAF protein. Vemurafenib also forms three hydrogen bonds with the following amino acid residues: Q530, D594 and G596. The compounds **12e**, **12i** and **12l** form additional hydrogen bonds with S536, G534 and S536, respectively. The compounds **12e** and **12i** both have additional hydrophobic interactions with D594, F468, G464 and S465, while the compound **12i** has hydrophobic interaction with Q461 and also with F468. The reference ligand (vemurafenib) demonstrated unique hydrophobic interaction with F516, F595, G593, I527, L505, S536, T529 and V482.

Molecular dynamics simulations of the interaction of the selected compounds with the V600EBRAF protein demonstrated that all four compounds maintained relative stability (RMSD values < 0.2 nm, Figure 5), with the reference ligand (vemurafenib) being the most stable among the four studied compounds.

On the basis of both MMGBSA and MMPBSA calculations (Figure 6), we found that the reference ligand demonstrated the lowest values of binding energies (−55 and −48 kcal/mol, respectively) among the four studied compounds. All of the three studied compounds showed very close values of binding energies against the binding site of the V600EBRAF protein, with the compound **12l** having slightly lower binding energies and the compound **12e** having slightly higher binding energies.

## 3. Conclusions

A novel set of 4-(1*H*-benzo[*d*]imidazol-1-yl)pyrimidin-2-amine linked sulfonamide derivatives **12a–n** was designed, synthesized, and tested over V600EBRAF. **12e**, **12i** and **12l** exhibited the highest percent inhibitions at 1µM fixed concentration. They were further tested over wild-type BRAF, V600EBRAF and CRAF to detect their selectivity and potency. They showed higher activity toward V600EBRAF compared to wild-type BRAF and CRAF. **12l** showed the dominant activity with an IC_50_ value over V600EBRAF of 0.49 µM, followed by **12i** and **12e** with IC_50_ of 0.53 µM and 0.62 µM, respectively. They were tested over NCI 60 cell line panel, where **12e** showed a wide range of activity against multiple cell lines. Heat map analysis revealed that the most sensitive cell line is SK-MEL-5, where **12l**, **12i** and **12e** showed the highest activity on both SK-MEL-5 and A375 cell lines (Table 4). The cell cycle analysis of **12e** showed an accumulation in both G1 and S phases.

## 4. Materials and Methods

### 4.1. Chemistry

Electrothermal Capillary apparatus (Staffordshire, UK) was used to detect all melting points. NMR spectral analyses were performed using a Bruker Avance 300 or 400 spectrometer ((Bruker Bioscience, Billerica, MA, USA) using TMS as standard, and chemical shift values were recorded in ppm. LC–MS analysis was performed using a Waters 3100 mass detector (Milford, USA), Waters 2998 photodiode array detector, Waters SFO system fluidics organizer, Waters 2545 binary gradient module, Waters reagent manager, Waters 2767 sample manager and Sunfire™ C18 column (4.6 × 50 mm, 5 μm particle size); solvent gradient = 95% A at 0min, 1% A at 5min; solvent A: 0.035% trifluoroacetic acid in deionized water; solvent B: 0.035% trifluoroacetic acid in methanol; FR = 3.0 mL/min. The AUC was calculated using Waters MassLynx 4.1 software. All reagents and solvents were purchased from Aldrich chemical Co. and Tokyo Chemical Industry (TCI) Co., and used without further purification.

#### 4.1.1. Synthesis of 1H-benzo[d]imidazole (**2**)

It was synthesized from *o*-phenylenediamine according to reported procedure [50].

#### 4.1.2. Synthesis of 1-(2-(methylthio)pyrimidin-4-yl)-1H-benzo[d]imidazole (**3**)

To a solution of **2** (10.1 g, 84.8 mmol) in DMF (30 mL), we added sodium hydride (6.00 g of a 60% suspension in mineral oil, 90.1 mmol) under N_2_ atmosphere and stirred the mixture for 1h. 4-Chloro-2-(methylthio)pyrimidine (16 g, 99.4 mmol) in DMF was slowly added, and the reaction mixture was stirred for an additional 12h at 80 °C, before being terminated with 10% NaHCO_3_, followed by extraction with methylene chloride. The organic layer was washed with water and dried using Na_2_SO_4_. The solvent was removed under reduced pressure to afford **3** as viscous oil (77%); ^1^H-NMR (400 MHz, DMSO-*d_6_*) δ 9.13 (s, 1H), 8.73 (d, 1H, *J* = 5.5 Hz), 8.41 (d, 1H, *J* = 8.0 Hz), 7.76 (d, 2H, *J* = 5.5 Hz), 7.45–7.35 (m, 2H), 2.64 (s, 3H). ^13^C-NMR (100MHz, DMSO-*d_6_*) δ 172.7, 159.8, 156.5, 144.9, 142.7, 131.7, 125.1, 124.4, 120.6, 115.6, 107.7, 105.6, 14.3. LC–MS (*m/z*) for C_12_H_10_N_4_S: 242.06, found: 243.11 (M+1)^+^.

#### 4.1.3. Synthesis of 1-(2-(methyl sulfonyl) pyrimidin-4-yl)-1H-benzo[d]imidazole (**4**)

A solution of potassium peroxymonosulfate (84.5 g, 124 mmol) in water (150 mL) was added dropwise to a methanolic solution of compound **3** (10.2 g, 41.3 mmol) in methanol (120 mL) at 25 degrees and stirred for an additional 48h. The mixture was reduced by removing methanolusing reduced pressure and extracted with methylene chloride (20mL). The organic layer was dried using Na_2_SO_4_ and evaporated under reduced pressure. The residue was purified by column chromatography (hexane/ethyl acetate, 2:1) to give **4** as a white solid (81%), m.p.: 180–182 °C. ^1^H-NMR (300 MHz, DMSO-*d_6_*) δ 9.35 (d, 1H, *J* = 6.0 Hz), 9.25 (d, 1H, *J* = 3.0 Hz), 8.52 (d, 1H, *J* = 9.0 Hz,), 8.01 (d,1H, *J* = 6.0 Hz), 7.82 (d, 1H, *J* = 9.0 Hz,), 7.53–7.39 (m, 2H), 3.55 (s, 3H). ^13^C-NMR (75 MHz, DMSO-*d_6_*) δ. 166.8, 164.1, 156.6, 145.0, 142.8, 131.7, 125.4, 124.6, 120.7, 115.8, 113.9, 39.9. LC–MS (*m/z*) for C_12_H_10_N_4_O_2_S: 274.05, found: 275.10; (M+1)^+^.

#### 4.1.4. Synthesis of Sustituted Ethanediamine and Propanediamine (**11a–n**)

Synthesis of side chains **11a–n** was carried out according to the reported literature [48].

#### 4.1.5. Synthesis of Target Compounds (**12a–n**)

To a solution of compound **4** (0.37 mmol) in DMSO (5.0 mL), DIPEA (0.6 g, 3.6 mmol) and **11a–n** (0.55 mmol) were added. The mixture was heated at 90 °C and stirred for 8h; then, the mixture was partitioned between ethyl acetate and water. The organic layer was dried using Na_2_SO_4_ and evaporated. The residue was purified by column chromatography (hexane/ethyl acetate, 2:1) to yield the desired products.

*N*-(2-((4-(1*H*-benzo [*d*]imidazol-1-yl) pyrimidin-2-yl) amino)ethyl)benzenesulfonamide **(12a)**: buff solid, (71%), m.p. 78–80 °C. ^1^H-NMR (400 MHz, CD_3_OD) δ 9.05 (s, 1H), 8.55 (d, 1H, *J* = 7.5 Hz), 8.08 (s, 1H), 7.86–7.81 (m, 3H), 7.72 (d, 1H, *J* = 7.5 Hz), 7.63–7.48 (m, 4H), 7.41–7.35 (m, 2H), 6.36 (d, 1H, *J* = 6.0 Hz), 3.62 (s, 2H), 3.18 (t, *J* = 6.5 Hz, 2H). ^13^C NMR (100 MHz, CD_3_OD) δ 155.6, 143.7, 142.0, 140.4, 132.3, 132.2, 131.7, 128.9, 128.8, 126.53, 126.49, 124.6, 123.3, 118.8, 115.7. LC–MS (*m/z*) for C_19_H_18_N_6_O_2_S: 394.12, found: 395.21 (M+1)^+^.

*N*-(2-((4-(1*H*-benzo [*d*]imidazol-1-yl) pyrimidin-2-yl) amino)ethyl)-4-fluorobenzenesulfonamide **(12b)**: white solid (67%), m.p. 74–76 °C. ^1^H-NMR (400 MHz, DMSO-*d_6_*) δ 9.06 (s, 1H), 8.50 (d, 1H, *J*= 7.5 Hz), 8.13 (d, 1H, *J*= 6.0 Hz), 7.96 (s, 1H), 7.88–7.80 (m, 2H), 7.76 (d, 1H, *J*= 7.5 Hz), 7.62–7.53 (m, 1H), 7.41–7.31 (m, 3H), 6.42 (d,1H, *J*= 6.0 Hz), 3.54 (d, 2H, *J*= 5.5 Hz), 3.04 (s, 2H), ^13^C-NMR (100 MHz, DMSO-*d_6_*) δ 165.8, 163.3, 155.7, 155.2, 145.0, 142.6, 137.2, 132.1, 129.9, 126.9, 124.6, 123.6, 120.2, 116.9, 104.3, 41.9, 40.7. LC–MS (*m/z*) for C_19_H_17_FN_6_O_2_S: 412.11, found: 413.21 (M+1)^+^.

*N*-(2-((4-(1*H*-benzo [*d*]imidazol-1-yl) pyrimidin-2-yl) amino)ethyl)-4-chlorobenzenesulfonamide **(12c)**: light yellow solid (77%), m.p. 82–84 °C. ^1^H-NMR (400 MHz, DMSO-*d_6_*) δ 9.05 (s, 1H), 8.55 (d,1H, *J* =7.5 Hz), 8.08 (d, 1H, *J* = 3.5 Hz), 7.75 (d, 1H, *J* = 7.5 Hz), 7.65–7.44 (m, 6H), 7.42–7.34 (m, 2H), 6.72 (d, 1H, *J* = 6.0 Hz), 3.63 (s, 2H), 3.18 (t, 2H, *J* = 6.5 Hz). ^13^C-NMR (100 MHz, DMSO-*d_6_*) δ 155.6, 143.7, 142.0, 140.4, 132.3, 132.2, 131.7, 128.9, 126.6, 124.3, 123.3, 118.8, 115.7, 41.4, 40.4. LC–MS (*m/z*) for C_19_H_17_ClN_6_O_2_S: 428.08, found: 429.20 (M+1)^+^.

*N*-(2-((4-(1*H*-benzo [*d*]imidazol-1-yl) pyrimidin-2-yl) amino)ethyl)-4-bromobenzenesulfonamide **(12d)**: light brown solid (70%), m.p. 80–82 °C. ^1^H-NMR (400 MHz, DMSO-*d_6_*) δ 9.07 (s, 1H), 8.51 (d, 1H, *J* = 8.0 Hz), 8.12 (d, 1H, *J* = 6.0 Hz), 7.97 (s, 1H, replaceable), 7.91 (s, 1H), 7.82 (d, 2H, *J* = 6.5 Hz), 7.77–7.73 (m, 3H), 7.61–7.55 (m, 3H), 7.37–7.31 (m, 1H), 6.43 (d, 1H, *J* = 6.0 Hz), 3.55 (d, 2H, *J* = 6.0 Hz), 3.03 (d, 2H, *J* = 5.5 Hz). ^13^C NMR (100 MH, DMSO-*d_6_*) δ 163.3, 155.8, 155.2, 145.0, 142.6, 140.6, 132.9, 132.1, 129.7, 126.9, 124.6, 123.6, 120.2, 116.0, 104.3, 42.6, 42.0. LC–MS (*m/z*) for C_19_H_17_BrN_6_O_2_S: 472.03, found: 473.10 (M+1)^+^.

*N*-(2-((4-(1*H*-benzo [*d*]imidazol-1-yl) pyrimidin-2-yl) amino)ethyl)-4-(trifluoromethyl)benzenesulfonamide **(12e)**: white solid (62%), m.p. 70–72 °C. ^1^H-NMR (400 MHz, CD_3_OD) δ 9.02 (s, 1H), 8.55 (d,1H, *J*= 7.5 Hz), 8.06–7.99 (m, 4H), 7.90 (d, 2H, *J*= 8.0 Hz), 7.84 (d, 2H, *J*= 7.0 Hz), 7.72 (d, 1H, *J*= 7.5 Hz), 7.41–7.32 (m, 2H), 6.33 (d, 1H, *J*= 6.0 Hz), 3.56 (s, 2H), 3.26 (d, 2H, *J*= 7.0 Hz). ^13^C-NMR (100 MHz, CD_3_OD) δ 162.6, 155.6, 144.4, 143.6, 141.9, 133.4, 131.7, 127.4, 126.0, 124.9, 124.3, 123.3, 118.8, 115.7, 40.2, 34.8. LC–MS (*m/z*) for C_20_H_17_F_3_N_6_O_2_S: 462.10, found: 463.10 (M+1)^+^.

*N*-(2-((4-(1*H*-benzo [*d*]imidazol-1-yl) pyrimidin-2-yl) amino)ethyl)-4-methoxy benzenesulfonamide **(12f)**: white solid (81%), m.p. 64–66 °C, ^1^H-NMR (400 MHz, DMSO-*d_6_*) δ 9.06 (s, 1H), 8.50 (d, 1H, *J* = 7.5 Hz), 8.12 (d, 1H, *J* = 5.0 Hz), 7.94 (s, 1H), 7.77–7.72 (m, 4H), 7.41–7.31 (m, 2H), 7.03 (d, 2H, *J* = 8.0 Hz), 6.42 (d, 1H, *J* = 6.0 Hz), 3.77 (s, 3H), 3.44 (d, 2H, *J* = 4.0 Hz), 3.00 (s, 2H). ^13^C-NMR (100 MHz, DMSO-*d_6_*) δ.162.4, 155.8, 154.9, 144.7, 142.6, 132.4, 132.1, 129.1, 124.6, 123.2, 119.9, 114.7, 104.3, 56.2, 42.1, 41.0. LC–MS (*m/z*) for C_20_H_20_N_6_O_3_S: 424.13, found: 425.20 (M+1)^+^.

*N*-(2-((4-(1*H*-benzo [*d*]imidazol-1-yl) pyrimidin-2-yl) amino) ethyl)-4-methyl benzenesulfonamide **(12g)**: yellow solid (52%), m.p. 66–68 °C. ^1^H-NMR (400 MHz, CD_3_OD) δ 9.02 (s, 1H), 8.52 (d1H, 7.0 Hz), 8.06 (s,1H), 7.80–7.69 (m, 4H), 7.41–7.33 (m, 3H), 7.23 (d, 2H, *J* = 8.0 Hz), 6.33 (d, 1H, *J* = 6.0 Hz), 3.59 (s, 2H), 3.17 (t, 2H, *J* = 6.5 Hz), 2.29 (s, 3H). ^13^C NMR (100 MHz, CD_3_OD) δ. 163.3, 155.5, 154.4, 143.2, 137.4, 131.7, 129.3, 126.6, 124.3, 123.3, 118.8, 115.7, 44.7, 41.4, 20.0. LC–MS (*m/z*) for C_20_H_20_N_6_O_2_S: 408.13, found: 409.18 (M+1)^+^.

*N*-(3-((4-(1*H*-benzo [*d*]imidazol-1-yl) pyrimidin-2-yl) amino)propyl)benzenesulfonamide **(12h)**: yellow solid (64%), m.p. 75–77 °C, ^1^H-NMR (400 MHz, DMSO-*d_6_*) δ 9.05 (s, 1H), 8.52 (d, 1H, *J* = 7.5 Hz), 8.10 (d, 1H, *J* = 5.0 Hz), 7.90 (s, 1H), 7.81–7.77 (m, 4H), 7.63–7.60 (m, 3H), 7.40–7.32 (m, 2H), 6.41 (d, 1H, *J* = 5.5 Hz), 3.07 (d, 2H, *J* = 6.0 Hz), 2.76 (t, 2H, *J* = 7.0 Hz), 1.54 (t, 2H, *J* = 7.0 Hz). ^13^C-NMR (100 MHz, DMSO-*d_6_*) δ 163.3, 161.5, 155.8, 155.0, 145.0, 142.6, 140.9, 132.8, 132.2, 129.7, 126.9, 124.5, 123.5, 120.2, 116.0, 104.1, 38.2, 35.1, 29.7. LC–MS (*m/z*) for C_20_H_20_N_6_O_2_S: 408.13, found: 409.20 (M+1)^+^.

*N*-(3-((4-(1*H*-benzo [*d*]imidazol-1-yl) pyrimidin-2-yl) amino)propyl)-4-fluorobenzenesulfonamide **(12i)**: white solid, Yield 59%, mp 80–82 °C, ^1^H NMR (400 MHz, DMSO-*d_6_*) δ 9.05 (s, 1H), 8.52 (d,1H, *J* = 7.24 Hz), 8.11 (d, 1H, *J* = 4.88 Hz), 7.83 (s, 3H), 7.76 (d, 2H, *J* = 7.56 Hz), 7.40–7.32 (m, 4H), 6.41 (d, 1H, *J* = 5.76 Hz), 3.46 (d, 2H, *J* = 4.16 Hz), 2.90 (d, 2H, *J* = 6.16 Hz), 1.76 (d, 2H, *J* = 7.24 Hz). ^13^C NMR (100 MHz, DMSO-*d_6_*) δ 163.33, 155.82, 155.04, 144.97, 142.56, 137.25, 132.14, 129.93, 124.52, 123.53, 120.24, 116.82, 115.96, 104.05, 40.99, 38.16, 29.17. LC–MS (*m/z*) for C_20_H_19_FN_6_O_2_S: 426.12, found: 427.21 (M+1)^+^.

*N*-(3-((4-(1*H*-benzo [*d*]imidazol-1-yl) pyrimidin-2-yl) amino)propyl)-4-chloro benzenesulfonamide **(12j)**: viscous oil (59%). ^1^H-NMR (400 MHz, CD_3_OD) δ 9.04 (s, 1H), 8.59 (d, 1H, *J* = 7.5 Hz), 8.08 (s, 1H), 7.81 (d, 2H, *J* = 7.0 Hz), 7.74 (d*,* 1H, *J* = 7.5 Hz), 7.64–7.56 (m, 1H), 7.44–7.35 (m, 4H), 7.37 (d, 1H, *J* = 6.0 Hz), 3.57 (s, 2H), 3.02 (t, 2H, *J* = 7.0 Hz), 1.85 (t, 2H, *J* = 7.0 Hz). ^13^C-NMR (100 MHz, CD_3_OD) δ 164.6, 157.8, 155.9, 143.7, 141.4, 140.3, 131.9, 128.9, 126.8, 124.7, 123.3, 118.6, 116.1, 112.9, 103.7, 42.6, 40.4, 28.9. LC–MS (*m/z*) for C_20_H_19_ClN_6_O_2_S: 442.09, found: 443.20 (M+1)^+^.

*N*-(3-((4-(1*H*-benzo [*d*]imidazol-1-yl) pyrimidin-2-yl) amino)propyl)- 4-bromo benzenesulfonamide **(12k)**: light brown solid (64%), m.p. 90–92 °C. ^1^H-NMR (400 MHz, DMSO-*d_6_*) δ 9.05 (s, 1H), 8.52 (d*,* 1H, *J* = 8.0 Hz), 8.10 (d, 1H, *J*= 6.0 Hz), 7.89 (s, 1H), 7.78–7.75 (m, 3H), 7.70 (s, 1H), 7.758–7.49 (m, 2H), 7.41–7.36 (m, 2H), 6.40 (d, 1H, *J* = 6.0 Hz), 3.46 (d, 2H, *J*= 6.0 Hz), 2.90 (d, 2H, *J* = 5.0 Hz), 1.74 (t, 2H, *J* = 7.0 Hz). ^13^C-NMR (100 MHz, DMSO-*d_6_*) δ 163.3, 155.8, 155.0, 145.0, 142.6, 140.8, 132.7, 132.2, 129.6, 126.9, 124.6, 123.5, 120.3, 116.0, 104.1, 40.9, 38.2, 29.2. LC–MS (*m/z*) for C_20_H_19_BrN_6_O_2_S: 486.04, found: 487.20 (M+1)^+^.

*N*-(3-((4-(1*H*-benzo [*d*]imidazol-1-yl) pyrimidin-2-yl) amino)propyl)-4-(trifluoromethyl) benzenesulfonamide **(12l)**: viscous oil (69%). ^1^H-NMR (400 MHz, CD_3_OD) δ 9.02 (s, 1H), 8.55 (d, 1H, *J* = 7.5 Hz), 8.06–8.01 (m, 3H), 7.90 (d, 2H, *J* = 8.5 Hz), 7.79 (s, 1H), 7.77 (s, 1H), 7.72 (d, *J* = 7.5 Hz), 7.41–7.32 (m, 2H), 6.33 (d, 1H, *J* = 6.0 Hz), 3.56 (s, 2H), 3.24 (t, 2H, *J* = 6.5 Hz), 1.69 (t, 2H, *J* = 7.0 Hz). ^13^C-NMR (100 MHz, CD_3_OD) δ 163.4, 162.6, 155.6, 144.4, 143.6, 141.8, 131.7, 127.4, 125.4, 124.3, 123.3, 118.8, 115.7, 40.2, 34.8, 29.2. LC–MS (*m/z*) for C_21_H_19_F_3_N_6_O_2_S: 476.12, found: 477.20 (M+1)^+^.

*N*-(3-((4-(1*H*-benzo [*d*]imidazol-1-yl) pyrimidin-2-yl) amino)propyl)-4-methoxy benzenesulfonamide **(12m)**: viscous oil (70%). ^1^H-NMR (400 MHz, CD_3_OD) δ 8.89 (s, 1H), 8.40 (d,1H, *J* = 5.0 Hz), 7.90 (s, 1H), 7.71–7.63 (m, 3H), 7.30–7.24 (m, 2H), 7.01–6.97 (m, 1H),6.83(d, 2H, *J* = 7.0 Hz), 6.20 (d, 1H, *J* = 4.0 Hz), 3.66 (s, 3H), 3.41 (s, 2H), 2.96 (t, 2H, *J* = 7.0 Hz), 1.77 (p, 2H, *J* = 7.0 Hz). ^13^C-NMR (100 MHz, CD_3_OD) δ 163.2, 162.8, 155.4, 154.0, 143.6, 141.8, 131.6, 128.7, 124.2, 123.2, 118.8, 115.8, 113.8, 103.5, 54.7, 40.7, 37.9, 28.7. LC–MS (*m/z*) for C_21_H_22_N_6_O_3_S: 438.14, found: 439.11 (M+1)^+^.

*N*-(3-((4-(1*H*-benzo [*d*]imidazol-1-yl) pyrimidin-2-yl) amino) propyl)naphthalene-1-sulfonamide **(12n)**: white solid (72%), m.p. 85–87 °C. ^1^H-NMR (400 MHz, DMSO-*d_6_*) δ 9.02 (s, 1H), 8.49 (d, 1H, *J* = 6.5 Hz), 8.42 (s, 1H), 8.07 (s, 4H), 7.95 (d, 1H, *J* = 7.0 Hz), 7.86–7.80 (m, 3H), 7.75 (d, 1H, *J* = 7.0 Hz), 7.67–7.59 (m, 2H), 7.33 (d, 2H, *J* = 5.5 Hz), 6.36 (d, 1H, *J* = 5.0 Hz), 3.44 (s, 2H), 2.95 (s, 2H), 1.75 (s, 2H). ^13^C-NMR (100 MHz, DMSO-*d_6_*) δ 156.0, 155.2, 145.0, 142.5, 137.9, 134.5, 132.1, 129.8, 129.5, 129.0, 128.2, 128.0, 128.0, 124.5, 123.5, 122.7, 120.2, 116.0, 104.0, 41.1, 38.2, 29.2. LC–MS (*m/z*) for C_24_H_22_N_6_O_2_S: 458.15, found: 459.20 (M+1)^+^.

### 4.2. In Vitro Anticancer Activity

#### 4.2.1. Anticancer Screening over 60 Cell Lines

Anticancer profiling over a panel of 60 cancer cell lines was carried out at the National Cancer Institute (NCI), Bethesda, Maryland, USA, https://dtp.cancer.gov/discovery_development/nci-60/methodology.htm (Last visit 22 September 2021), using the standard procedure (detailed procedure is located in the Appendix A page 50 s).

#### 4.2.2. Antiproliferative Effect over A375 and SK-MEL-5

The effect of final target compounds over A375 and SK-MEL-5 was performed using reported procedures [48] (detailed procedure is located in the Appendix A page 51s).

### 4.3. Enzyme Assay

The enzymatic assays were performed in Reaction Biology Corp. using the standard protocol at 1 µM ATP and threefold dilution factor (detailed procedure is found in the Appendix A page 52 s).

### 4.4. Cell Cycle Analysis

In the beginning, the IC_50_ of compound 12e over T-47D (American Type Culture Collection, US) was determined using MTT assay, and the same procedure was applied for A375 and SK-MEL-5. T-47D cells (105 cells) were treated with **12e** (5 µM) and incubated for 48h. Cells were collected by trypsinization, washed two times with ice-cold PBS, and re-suspended in 60% cold ethanol and kept at 4 °C for 60 min. PBS was used to wash the fixed cells and was re-suspended in 1.0 mL of PBS containing 10µg/mL propidium iodide and 50 µg/mL RNAase A. Cells were incubated in the dark at 37 °C for 20min; then, DNA contents were analyzed using flow cytometric analysis using an FL2 (λex/em 535/617 nm) signal detector. For each sample, 12,000 events were acquired. The cell cycle distribution was obtained by using ACEA NovoExpress^TM^ software [51].

### 4.5. Molecular Docking

Crystal structure of the V600EBRAF protein in complex with vemurafenib (PLX-4032) was downloaded from the Protein Data Bank [52] (PDB, ID: 3OG7). The missing amino acid residues of the protein in the crystal structure were modelled using the CHARMM-GUI web-server [53]. Molecular docking, parametrization of the protein structure and studied compounds, visualization of the binding poses, and 2D interaction diagrams were carried out using ICM-PRO software [54]. Colors in 2D interaction diagrams (Figure 5) represent the following: green: hydrophobic region, blue: hydrogen bond acceptor, gray parabolas: accessible surface for large areas, and dashed arrows: hydrogen bonds. The mfScore of ICM-PRO software and MMPBSA method were used for the binding energy estimation. For MMPBSA calculations, the following script https://github.com/sahakyanhk/iPBSA (Last visit on 22 August 2021). which uses the ff14SB force field for protein parameterization, and General Amber Force Field (GAFF) with AM1-BCC charge model for small molecule parametrization, were used [55].

### 4.6. Molecular Dynamics Simulations

The AMBER 20 software package was used for the molecular dynamics simulation [56]. Protein parametrimization was performed using ff14SB force field, while general amber force field (GAFF) with AM1-BCC charge model was used for the ligand parameterization. Docked conformations of three studied compounds (**12i**, **12e** and **12l**) and reference ligand obtained from previous stage was used as a starting points for corresponding molecular dynamics simulations. Complexes of V600EBRAF (PDB, ID: 3OG7, chain A) with studied compounds and reference ligand were solvated in the TIP3P water model and Na+/Cl− ions at 150 mM concentration [57]. The Monte Carlo barostat (reference pressure at 1 bar) [58] and Langevin thermostat (collision frequency: gamma_ln 2 ps^−1^) [59] were used to keep the temperature at 310.15 K. We used a 1 nm cutoff for the long-range electrostatic interactions using The Particle Mesh Ewald (PME) method. The first 5 ns of all performed simulations were system minimization and equilibration steps. Each of the generated complexes underwent 100 ns of conventional molecular dynamics simulation. Finally, for every simulation, binding free energies were used, wherein binding affinity was calculated using MMGBSA and MMPBSA methods using MMPBSA.py program [60]. A total of 250 snapshots with equal intervals collected from the last 20 ns for every simulation were used for binding energy calculations.

## Data Availability

Not applicable.

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
