# Peer review of "Design, Synthesis and Anticancer Profile of New 4-(1H-benzo[d]imidazol-1-yl)pyrimidin-2-amine-Linked Sulfonamide Derivatives with V600EBRAF Inhibitory Effect"

_ijms, 2021, doi:10.3390/ijms221910491_

Round 1
Reviewer 1 Report
The work entitled “Design, Synthesis and Anticancer Profile of New 4-(1H-2 benzo[d]imidazol-1-yl)pyrimidin-2-amine Linked Sulfonamide 3 Derivatives with V600EBRAF Inhibitory Effect” describes the design and synthesis of a series of benzimidazole based compounds decorated with sulfonamide moiety, in order to affect V600EBRAF. The novel 14 derivatives have been evaluated through enzymatic as well as cellular assays in order to establish their effectiveness as anticancer agents. Molecular docking as well as molecular dynamic studies have also been performed.
However, the manuscript needs of some important revisions, including an improvement of English language and style.
- In all the reported assays, i.e. enzymatic as well as cellular ones, there is the lack of a reference drug, that has to be included.
- The discussion of the results obtained from enzymatic as well as cellular assays has to be ameliorated. Currently these parts only report the % values of inhibition or the IC50 values. Instead, these data should be correlated with structural properties of the compounds in order to obtain SAR description.
- The experimental part lack of information about the anticancer screening on 60 cell lines. Albeit performed by NCI this information has to be included. Furthermore, the link included does not work. What is the compounds concentration in the assay performed on cancer cell lines?
- Why for the cell cycle analysis has been employed the concentration of 5 µM and what is the reason of the choice of this cell line? Please add the info and also improve the discussion of the results. Are these changes related to the blockage of V600EBRAF? There are other putative mechanisms?
- The quality of the images reported in the figure 3 has to be improved.
In the text there are some grammatical error and typos that should be corrected:
- Please check all the IUPAC names reported in the ms. For example, in line 78 please change H in the italic style in the name 1-(2-(methylsulfonyl)pyrimidin-4-yl)-1H-benzo[d]imidazole.
- Line 22, please add the space to 1µM
- Line 41, please correct singling;
- Scheme 2, please add the point after 11a-n.
Author Response
Dear Respected Reviewer
Thank you for your highly valuable comments and here in our responses to these comments.
comment: In all the reported assays, i.e. enzymatic as well as cellular ones, there is a lack of a reference drug that has to be included.
response: we have added Sorafenib as the positive standard and its data for both enzymatic assay and cellular assay.
comment: The discussion of the results obtained from enzymatic as well as cellular assays has to be ameliorated. Currently, these parts only report the % values of inhibition or the IC50 values. Instead, these data should be correlated with the structural properties of the compounds in order to obtain SAR description.
response: A paragraph that correlates structural properties to activity is present on page 6 at the end of the enzyme assay part.
comment: The experimental part lack information about the anticancer screening on 60 cell lines. Albeit performed by NCI this information has to be included. Furthermore, the link included does not work. What is the compounds concentration in the assay performed on cancer cell lines?
Response: The link was updated and a detailed procedure was added in supporting information.
comment: Why for the cell cycle analysis has been employed the concentration of 5 µM and what is the reason for the choice of this cell line? Please add the info and also improve the discussion of the results. Are these changes related to the blockage of V600EBRAF? There are other putative mechanisms?
responses: Information were added
comment: The quality of the images reported in figure 3 has to be improved.
response: Figure 3 was removed and added to supporting information to be in high-resolution formate
comment: In the text, there are some grammatical errors and typos that should be corrected:
responses: The whole manuscript text was re-checked and errors were corrected
comment: Please check all the IUPAC names reported in the ms. For example, in line 78 please change H in the italic style in the name 1-(2-(methylsulfonyl)pyrimidin-4-yl)-1H-benzo[d]imidazole.
response: All compounds were checked and the IUPAC name was corrected for each compound
comment: Line 22, please add the space to 1µM
response: Done
comment: Line 41, please correct singling;
response: Done
comment: Scheme 2, please add the point after 11a-n
response: Done
Reviewer 2 Report
The manuscript “Design, Synthesis and Anticancer Profile of New 4-(1H-2 benzo[d]imidazol-1-yl)pyrimidin-2-amine Linked Sulfonamide 3 Derivatives with V600EBRAF Inhibitory Effect” of Abdel-Maksoud et al. is well written and of great interest. Work showed the design of sulfonamides inhibitor against the mutant kinase V600EBRAF, an important target against melanomas. The inhibitory profiles of the synthesized compounds were evaluated against V600EBRAF (compared with the wild type kinase BRAF and kinase CRAF) as well as their cytotoxicity against nine different cancer cell lines. Finally, three derivatives with the best inhibitory activity and vemurafenib were submitted to in silico studies (docking, MMPBSA, MMGBSA, and MD). I recommend this work for publication after minor revision.
- In molecular docking discussion is not immediately clear which are the important residues of V600EBRAF essential for its activity. Please define that.
- Moreover, the docking figures are too small and it is so many difficult to understand the interaction in the 2D schemes. Please enlarge them all. You could obtain 2 figures with 4 panels (e.g. Figure 5A: docking of 12e and 12i, with the respectively 2D schemes below; Figure 5B: docking of 12l and vemurafenib, with the respectively 2D schemes below), or a big one that is the fusion of Figure 5A and 5B (e.g. Figure 5: docking of 12e and 12i, with the respectively 2D schemes below; docking of 12l and vemurafenib below, with the respectively 2D schemes below).
Author Response
Dear respected reviewer
Thank you for your highly valuable comments, herein, you can find our responses to these comments
comment: In molecular docking, discussion is not immediately clear which are the important residues of V600EBRAF essential for its activity. Please define that.
response: the discussion part has been updated to be clear enough
comment: The docking figures are too small and it is so many difficult to understand the interaction in the 2D schemes. Please enlarge them all. You could obtain 2 figures with 4 panels (e.g. Figure 5A: docking of 12e and 12i, with the respectively 2D schemes below; Figure 5B: docking of 12l and vemurafenib, with the respectively 2D schemes below), or a big one that is the fusion of Figure 5A and 5B (e.g. Figure 5: docking of 12e and 12i, with the respectively 2D schemes below; docking of 12l and vemurafenib below, with the respectively 2D schemes below)
response: the titled figure has been changed
Round 2
Reviewer 1 Report
Dear Authors,
I appreciate the efforts and the improvements of the manuscript, however there are some issues that have not be solved.
- The supporting info file doesn’t not report the procedure about the anticancer screening (for example: what is the concentration of the inhibitors in these experiments? What is the media for the cell growth? What is the solvent used to dissolve the inhibitors? ) and evaluation of IC50 on the cell lines.
- The supporting info file doesn’t not report the heat map analysis.
- In the reply the Authors assessed that the 5 µM concentration has been chosen based on the IC50 value of the compound 12e against T-47D. However, IC50 values have been reported in the manuscript only for the SK-MEL-5 and A375 cell lines (Table 3). How the Authors evaluate the IC50 against T-47D cell lines? Please add data and experimental procedure.
- The experimental procedure of the enzymatic assays has to be added.
- The experimental procedure about IC50 evaluation on SK-MEL-5 and A375 cell lines has to be reported.
- The new link for the experimental procedure doesn’t work, too.
Other suggestions:
In the IUPAC name 4-(1H-benzo[d]imidazol-1-yl)pyrimidin-2-amine please correct 1.
In the figure 2 please refer to the supporting info file to see the cell lines type employed in the screening
Author Response
Dear respected reviewer
Thank you for your invaluable comments, kindly find our responses to your comments
Comment # 1: The supporting info file doesn’t not report the procedure about the anticancer screening (for example: what is the concentration of the inhibitors in these experiments? What is the media for the cell growth? What is the solvent used to dissolve the inhibitors? ) and evaluation of IC50 on the cell lines.
- Response: Complete procedure for NCI assay was added to supporting information. And each question answer was highlighted in page 50s
Comment # 2: The supporting info file does not report the heat map analysis.
- Response: Heat map analysis figures are located on page 3s and 4s
Comment # 3: In the reply, the Authors assessed that the 5 µM concentration has been chosen based on the IC50 value of the compound 12e against T-47D. However, IC50 values have been reported in the manuscript only for the SK-MEL-5 and A375 cell lines (Table 3). How do the Authors evaluate the IC50 against T-47D cell lines? Please add data and experimental procedure.
- Response: Determination of IC50 of compound 12e over T-47D is a standard protocol on cell cycle to detect the used concentration. The procedure used was the same as SK-MEL-5 and A375 ( MTT assay) and this information was added to exp. section
Comment # 4: The experimental procedure of the enzymatic assays has to be added.
- Response: Enzyme assay procedure was added to supporting information Page 52s.
Comment # 5: The experimental procedure for IC50 evaluation on SK-MEL-5 and A375 cell lines has to be reported.
- Response: The detailed procedure was added to supporting information page 51s.
Comment # 6: The new link for the experimental procedure doesn’t work, too
- Response: The link was checked (please hold ctrl and double click the link.)
Comment # 7: Other suggestions: In the IUPAC name 4-(1H-benzo[d]imidazol-1-yl)pyrimidin-2-amine please correct 1.
- Response: Correction was performed.
Comment # 7: In figure 2 please refer to the supporting info file to see the cell lines type employed in the screening
- Response: Refereeing to supporting information was done
Round 3
Reviewer 1 Report
Please correct IC50 and CO2 in the supporting information part (subscript)